# Effect of Mineral Admixtures on the Sulfate Resistance of High-Strength Piles Mortar

**DOI:** 10.3390/ma13163500

**Published:** 2020-08-08

**Authors:** Yanyan Hu, Linlin Ma, Tingshu He

**Affiliations:** 1College of Materials Science and Engineering, Xi’an University of Architecture and Technology, Xi’an 710055, China; mll2336@126.com (L.M.); hetingshu@xauat.edu.cn (T.H.); 2State Key Laboratory of Green Building in Western China, Xi’an University of Architecture and Technology, Xi’an 710055, China

**Keywords:** PHCP mortar, admixtures, pore structure, hydration products, sulfate corrosion resistance

## Abstract

Pre-stressed high-strength concrete piles (PHCP) are widely used in the building industry in China. The main aim of our research was to investigate the utilization of quartz powder, fly ash, and blast furnace slag as mineral additives to prepare PHCP mortar. The samples were prepared using steam and autoclaving steaming. The influence of minerals on the sulfate resistance of mortar was analyzed using X-ray diffraction (XRD), scanning electron microscopy (SEM) and mercury intrusion porosimetry (MIP) tests. The results showed that when compared to single doped quartz powder samples, samples prepared using fly ash or blast furnace slag improved the sulfate resistance of the PHCP mortar. Furthermore, the resistance to sulfate attack of samples with dual doped quartz powder, fly ash, and blast furnace slag also improved. MIP tests showed that mineral additives can change the pore size distribution after autoclave curing. However, the number of aching holes increased after mixing with 20% quartz powder and caused a decrease in the sulfate resistance.

## 1. Introduction

The pre-stressed high strength concrete pipe pile (PHCP) is widely used in the reinforcement of the soft soil foundation of buildings, bridges, ports, railways and wharf constructions in China due to its high strength, the convenience of construction and cost-saving, etc. [1,2,3].

PHCP is generally prepared by autoclaved curing [4,5] after atmospheric steam curing [6,7]. Elevated temperature can effectively improve the early stage hydration rate of the cement clinkers and the strength of PHCP (≥ 80 MPa) [8,9]. However, autoclaved curing may increase the capillary porosity [10,11] and reduce the gel porosity of the hydrated cement matrix, which leads to a deterioration in durability performances [12,13,14].

In order to avoid the negative effect of autoclave curing on the durability of PHCP, many scholars have tried to add mineral admixtures to PHCP, such as quartz powder (QP) [15,16], fly ash (FA) [17,18], blast furnace slag (SG) [6,19], and silica fume (SF) [20]. Under the action of hydrothermal synthesis, the activity of mineral admixtures in concrete is very well stimulated. QP [15,21] is the most important mineral admixture in PHCP in China. During steam curing at atmospheric pressure, SiO_2_ in QP is inactive and cannot react with Ca (OH)_2_ formed during the hydration of the cement [15]. Under high temperature and high pressure, QP can react with cement hydration products to form tobermorite crystals [22], which prevents the formation of high crystallinity phases and reduces the content of Ca (OH)_2_ crystals in the hydration products to improve the compressive strength of the concrete [23].

Hydrothermal synthesis reaction of QP and cement hydration products:(1)$$\mathrm{Ca}{\left(\mathrm{OH}\right)}_{2}+{\mathrm{SiO}}_{2}\overset{180–220\;{}^{\circ}C}{\to }{\;C}_{5}S_{6}H_{5}$$

An increasing amount of attention has been paid to the durability of mineral admixtures in PHCP. Alawad et al. [6], Alhozaimy et al. [7] reported that ground dune sand and SG in autoclave-cured cement mix produced a compacted microstructure. Xue et al. [24] found that the compressive strength of PHCP concrete with 20% ground silica sand or with 30% SG can reach more than 80 MPa after autoclaved curing; the PHCP concrete with 20% ground silica sand shows the best mechanical properties. Bahedh et al. [17], Wu [25], Tan and Zhu [26] studied the influences of FA, SG, and SF on the chloride permeability of autoclaved concrete. They found that the replacement of FA, SG, and silica fume dramatically decreased the chloride ion permeability of the autoclaved concrete, and FA was the most efficient means to reduce chloride permeability. However, Wei et al. [27], Xu et al. [28] reported that PHCP has poor frost resistance and sulfate resistance after adding QP.

Sulfate ions are abundant in the underground soil in the west and south of China [29,30,31]. Therefore, the purpose of this study is the sulfate resistances of PHCP mixed with QP, FA, and SG. The pore structure after autoclaved curing was also measured by the mercury intrusion method (MIP). Meanwhile, the hydration mechanisms and the morphology of hydration products were studied using X-ray diffraction (XRD) and scanning electron microscopy (SEM).

## 2. Materials and Methods

### 2.1. Raw Materials

The cement used for the experiments was ordinary Portland cement (OPC), with a strength grade of 42.5, which complies with the Chinese National Standard: GB/T 175-2007 [32]. The fineness was 301 m^2^/kg. The physical and mechanical properties are shown in Table 1. The quartz powder (QP) was acquired from Lehui Building Materials Co., Ltd. of Shanxi (Xi’ an, China), and had a fineness of 420 m^2^/kg. Blast-furnace slag (SG) was a granulated ground product, with a fineness of 468 m^2^/kg, provided by Delong Mineral Powder Co., Ltd. of Hancheng (China). The fly ash (FA) used in this research was class II based on the Chinese National Standard: GB/T 50146-2014 [33] with a fineness of 330 m^2^/kg. The chemical compositions of the raw materials are listed in Table 2.

The fine aggregate chosen was clean natural sand with a specific gravity of 2.61, a fineness modulus of 2.42 and a maximum size of 4.75 mm. The coarse aggregate used was (4.75–9.5) mm crushed granite stone, with a specific gravity of 2.69. The naphthalene superplasticizer was produced by Tongcheng Construction Technology Co., Ltd. of Xi’an (China) with a mixing amount of 1% for the cementing material, with 25.3% water-reducing rate. The water used in this experiment was tap water.

### 2.2. Specimen Preparation and Curing Conditions

The mass ratio of cementitious material to crushed stone was 1:0.35; the water-binder ratio was 0.29. The binder of mortar specimen with the size of 40 mm × 40 mm × 160 mm was prepared according to the Chinese National Standard: GB/T 17671-1999 [34]. The surfaces of the samples were covered with a plastic film in order to prevent water evaporation. The schematic representation of the process is shown in Figure 1.

### 2.3. Experimental Procedure

After the autoclaving process, the samples were divided into two groups. One group was put in a curing room at 20 ± 1 °C in water until testing. The other group was inserted into an NJW-LS (Nai Jiu Wang) computer control equipment to simulate the sulfate dry–wet cycle. The dry–wet cycles were carried out according to the Chinese National Standard: GB/T 50082-2009 [35]. That is, it was immersed in 5% Na_2_SO_4_ solution at room temperature for 16 h, aired for 1 h, dried for 6 h at 80 ± 5 °C, and then cooled for 1 h to test the compressive strength. One dry and wet cycle lasted for 24 h in total.

The sodium sulfate solution was 5%. The pH value of the solution was measured once every two weeks until the specimen was destroyed. After finishing the sulfate dry–wet cycles, the surfaces were cleaned with a soft brush, and then the specimens were observed. Every specimen was tested for compressive strength.

This test uses the evaluation index of relative strength and corrosion resistance coefficient. According to the following equation of the Chinese National Standard: GB/T 50082-2009 [35]:$$K_{n}=\frac{R_{2}}{R_{1}}\times 100$$

$K_{n}$ represents strength and corrosion resistance coefficient; the *n* in $K_{n}$ represents the number of dry–wet cycles for sulfate corrosion resistance; *R*_1_ represents the compressive strength of the standard cured samples; *R*_2_ represents the compressive strength of the samples under dry–wet cycles.

The microstructure images of the cement pastes and aggregate interfaces with different admixtures added after 300 sulfate dry–wet cycles were studied by Scanning Electron Microscopy (SEM), using the JXA-840A from Mettler Toledo (Columbus, OH, USA). A sample of the fractured center part of the autoclaved-curing cement stone was selected, crushed, and immersed in absolute ethanol to stop the hydration. A thin film with a thickness of 2 mm was cut from the erosion surface of the fractured center part of the autoclaved-curing cement stone sample using a cutter. The sample was treated with vacuum and gold spraying, and the test piece was attached to the copper production with a conductive adhesive.

The hydration product from the specimen after 300 sulfate dry–wet cycles was investigated by XRD using the Japan Rigaku D/Max 2400 (Japan Rigaku Co., Ltd, Tokyo, Japan). The center part of the broken specimen was ground finely. The fineness was controlled at an 8 mm square hole screen, and the screen residue did not exceed 10% to prepare the XRD sample.

The Micromeritics Auto-Pore 9200 of McMuritik Instrument Co., Ltd. (Shanghai, China) was used to analyze the pore structure of the autoclaved cement specimen. The samples were broken after the compressive strength test. Particles below 5 mm were screened out and immersed in absolute ethanol to stop hydration. After 48 h, they were taken out and placed in a vacuum drying oven at 60 °C, baked to constant weight, and then the MIP test was conducted.

## 3. Results and Discussion

### 3.1. Compressive Strength Corrosion Resistance Coefficient

The obtained results of $K_{n}$ coefficients are enlisted in Table 3, where OPC refers to the primary material, without mineral additives; QP20: 20% weight value of cement with QP; SG20, SG30: 20%, 30% weight value of cement with SG; FA10: 10% weight value of cement with FA; QP10SG20: 10% weight value of cement with QP and 20% weight value of cement with SG; QP10FA20: 10% weight value of cement with QP and 20% weight values of cement with FA.

Table 3 shows the effect of mineral additives on the sulfate resistance of PHCP. As it can be observed, the value of $K_{n}$ increases until K240. After 90 dry–wet cycles, the values K90 of SG30 and QP10SG20 reach their peaks, 14.0% and 15.0% higher than the standard specimens respectively. It can be supposed that the capillary pores were filled by hydration products produced by earlier sulfate erosion. Moreover, the compressive strength decreases with the increasing number of drying–watering cycles. After 150 cycles, the compressive strength of all samples decreased, reaching its maximum in the case of sample QP20. After 300 cycles, the sample SG20 was entirely destroyed, while the sample QP10SG20 had the highest corrosion resistance coefficient. The results indicate that adding QP and SG together can improve sulfate resistance.

### 3.2. Visual Observation

The samples after 300 dry–wet cycles of sulfate treatment are shown in Figure 2.

The sample SG20 showed the most severe corrosion damage, as it was entirely destroyed. The cement stone was loose and separated from the aggregate. Sample SG20 was nearly complete, with a small amount of corrosion pits on the surface. SG30 and FA10 did not have any serious/visible damage, as the degree of angular loss and epidermal shedding was lighter than that for sample QP20, where the cracks were visible, having been penetrated from the surface to the internal regions of the sample. The damages of QP10SG20 and QP10FA20 were similar to that of SG30, the samples being relatively complete, but one side of the vertical forming surface suffered degradation at the edges.

After soaking in Na_2_SO_4_ solution for a long time, the sample surfaces had high concentrations of sulfate. Then, gypsum crystallization/erosion occupied the dominant position. Having relatively low sulfate concentration in the internal part, ettringite crystallization erosion occupied the dominant position. There was no sulfate penetration in the core of the hardened cement paste. There was no gypsum and ettringite, the internal remained intact [36]. The destruction of the porous structure, and the growth of the ettringite crystal can lead to crystallization pressure and water absorption swelling, with fine needle-like and flaky crystals. The expansion of the internal volume stress in the paste causes the expansion of cracks in the cement-based material. On the other hand, crystallization of the gypsum can induce two adverse effects: (i) the volume of gypsum crystal increases by about 124%, causing the expansion and cracking of cement-based materials and (ii) the hydration product calcium hydroxide (CH) contributes to the mechanical properties of the hardened paste. As the reaction of sulfate and hydration product produce gypsum, the consumption of CH leads to a decrease in strength and durability, inducing the decrease of the mechanical properties of the paste, from lacking edge or even resulting in the collapse of the structure.

### 3.3. Microstructural Investigations by XRD and SEM

The XRD patterns of white crystals found on the surface of the samples, mixed with QP20 after 300 dry–wet cycles are shown in Figure 3, indicating that these white crystals are Na_2_SO_4_ crystals.

In order to obtain information from the analysis of the morphology, the samples aged for the same time, treated with the same standard curing, and 300 dry–wet cycles were analyzed. Their hydration products are shown in Figure 4.

In the alkaline environment of the hydration cement paste, the hydration of SG and the secondary hydration of FA can consume part of CH in the paste and reduce the alkalinity of the paste. The hydrothermal synthesis of QP and CH led to a decrease in CH content and gypsum crystal production. Meanwhile, due to the low water-binder ratio, the addition of mineral additives can refine the porosity of paste and can prevent the external sulfate ions from entering into the cement-based materials. Therefore, the amount of gypsum in Figure 4b was reduced.

According to the reaction of ettringite formation in the cement-based materials under sulfate attack [37], the amount of generated ettringite is controlled by the content of hydrate aluminate in the paste when there is enough CH and SO_4_^2−^. The content of the hydrate aluminate is related to the content of active Al^3+^. From Table 2, it can be observed that the content of active Al_2_O_3_ in QP, FA, and SG is 1.99%, 29.20%, and 14.64% respectively. The results are relatively higher for the latter two materials.

A large part of Al^3+^ in fly ash can be found in the inert mullite crystals, while Al^3+^ in slag mainly exists in the active amorphous phase. Moreover, if the hydration of cement produces sufficient alkali agents, the activity of SG is greatly stimulated.

Al^3+^ and AlO_4_^5−^ in the active intangible phase are more easily precipitated. Moreover, some of these ions are bound to the hydrated C-S-H gel structure, with a considerable portion of Al^3+^ and AlO_4_^5−^, which can enter the pores with the solution and become free ions or new hydrated aluminate phase combined with the hydration product of cement [38]. When the sample is immersed in the sodium sulfate solution, the free Al^3+^, AlO_4_^5−^ or hydrated aluminate phase can react with SO_4_^2−^ and form ettringite crystals. It can be seen that although the addition of SG can dilute the content of C_3_A in cement and reduce the alkalinity of cement paste after hydration, the slag itself can precipitate active Al^3+^ and AlO_4_^5−^ into the paste, which may accelerate the occurrence of ettringite erosion. The diffraction peak at about 9.1° corresponds to Aft (Ettringite). The peak is not observed in Figure 4. Ettringites mostly decompose during autoclaved curing. Therefore, as it can be seen in Figure 2c,d, the damage of the sample with SG is not so severe. It can be observed from Figure 4a, the CH diffraction peak of 20% QP and 10% FA were similar. However, Figure 2b,f,g show that the damage of 20% QP was the most serious. Therefore, the damage of paste mixed with 20% quartz powder cannot be explained from the information shown in Figure 4.

The cement pastes and interface zones of specimens after 300 dry–wet sulfate cycles were analyzed. The SEM micrographs of cement paste interfaces with different admixtures are shown in Figure 5.

Figure 5 shows that there were no cracks at the interface of OPC, FA10, and SG20, but many tiny cracks can be found at the interface of QP20. By magnifying these cracks, it can be observed that some small particles are condensed into clumps of crystals and sheets of material, and the interface is loose and porous, accelerating the destruction of the sample. When QP, SG, or FA was applied, particles of different sizes were dispersed in the paste. Some of them filled the pores, which reduced the pore size and porosity connectivity, inhibiting the invasion of external SO_4_^2−^. Moreover, most of these particles underwent a hydration reaction on the surface and consumed a part of CH. Secondary hydration products were also produced as the pore structure of the paste was refined by filling pores with these products and the sulfate corrosion resistance was improved.

### 3.4. Pore Structure Analysis

The influence of pore distribution can be more important than porosity on the macroscopic behavior of concrete [39]. Many studies on the microstructure of cement hydration with different mineral admixtures indicate that this can effectively improve the distribution of pores, optimize the structure of the pores, increase the content of gel in the pores, and reduce the number of capillary pores [40], improving in this way the durability of the concrete.

The results of the mercury intrusion method (MIP) are shown in Figure 6 and Figure 7. The percentages of pores with diameter < 0.02 μm to the total volumes of pores were: 61.1% in OPC, 66.6% in QP20, 70.2% in SG20, 63.2% in FA10, and 64.3% in QP10SG20, indicating that most of the pores in the high-pressure steam cement paste are harmless pores (gel pores). The analysis shows that under the condition of steam and autoclave based treatment, the hydration rate is increased, as the CH is consumed. More C-S-H and C-A-H are generated, resulting in an increase in gel holes. At the same time, when the pore diameter is less than 0.02 μm, the maximum hole aperture in OPC, QP20, SG30, FA10, and QP10SG20 was 16.4 nm, 12.1 nm, 9.5 nm, 12.7 nm, and 13.2 nm, respectively, showing that the pore size of autoclaved cement stone was refined after adding the mineral additives.

The pozzolanic reactivity, the filling and micro-aggregate effect of QP, and other additives can effectively improve the pore structure [41], inhibiting the formation of large-sized ettringites under sulfate attack [42]. Although the increase in the number of gel holes is beneficial to the improvement of concrete durability [43], high-temperature treatment can play a role in pore size distribution. It can be seen from Figure 7 that the number of harmful holes (> 0.2 μm) was higher than other samples. This can be attributed to the treatment, as the temperature gradient increases correspondingly; in parallel, the porosity and capillary content increases. Moreover, expansion stress leads to the occurrence and growth of cracks. The resultant dilatation pressure damages the pore structure inside the cement stone, so the resistance of concrete to sulfate attack is reduced.

## 4. Conclusions


(1)Compared with the blank specimen, the sulfate resistance of PHCP with mono doped QP decreased, but that of PHCP with mono dopes SG or FA increased. Furthermore, the sulfate attack of PHC with dual dopes QP, SG, and FA increased, respectively.(2)Adding mineral admixtures can decrease the size of pores. The reactive reaction, the filling and micro aggregate effect of QP and other additives can effectively improve the pore structure and increase the content of gel pores, inhibiting in this way the formation of large ettringite in the sulfate environment. The durability of autoclaved mortar was improved in this way.(3)Autoclaved curing changes pore size distribution and refine the aperture. However, the number of harmful holes mixed with QP20 increases, and sulfate resistance performance decreases.


## Figures and Tables

**Figure 1 materials-13-03500-f001:**
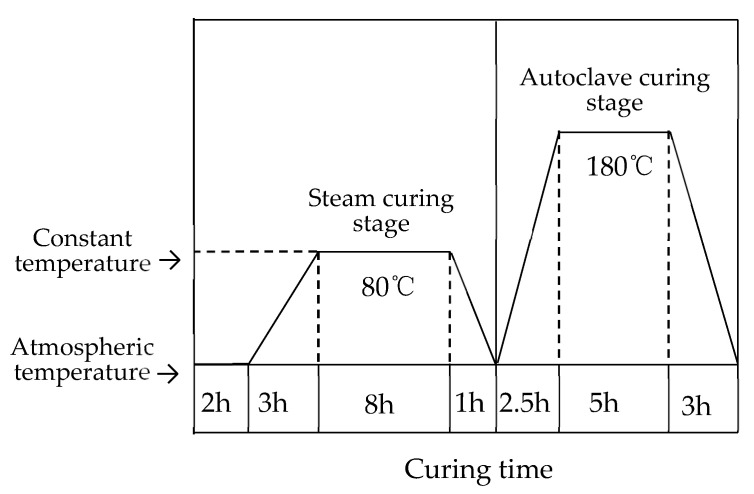
Steam curing and autoclaved curing.

**Figure 2 materials-13-03500-f002:**
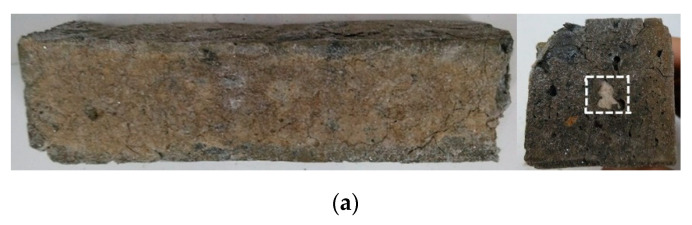
The samples added with different admixtures. After 300 dry–wet cycles of sulfate treatment. (**a**): OPC; (**b**): QP20; (**c**): SG20; (**d**): SG30; (**e**): QP10SG20; (**f**): FA10; (**g**): QP10FA20.

**Figure 3 materials-13-03500-f003:**
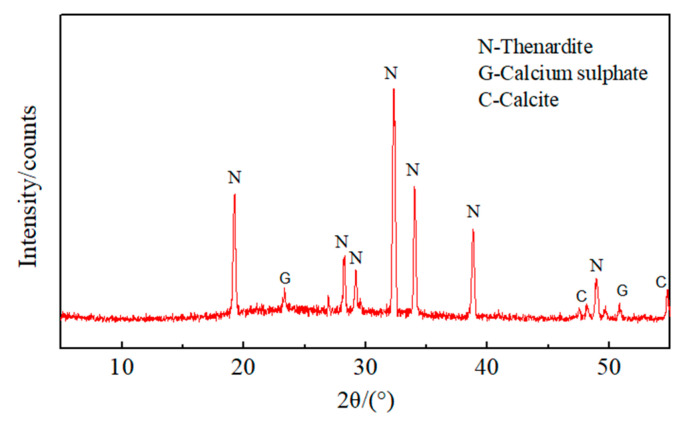
X-ray diffraction (XRD) diagram of white particles on the surface of QP20 sample. After 300 sulfate dry–wet cycles.

**Figure 4 materials-13-03500-f004:**
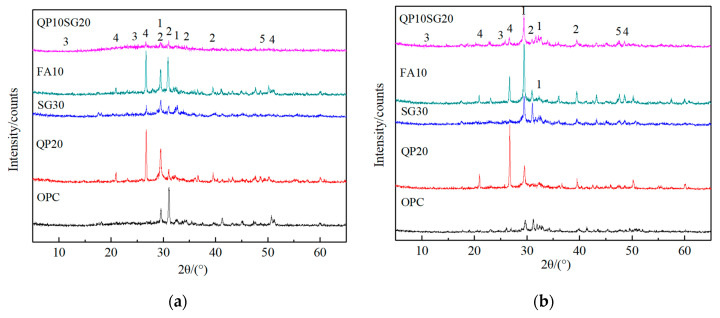
XRD diagram of the hydration product from the same age of the specimen after standard curing and 300 sulfate dry–wet cycles. (**a**) standard curing (**b**) 300 sulfate dry–wet cycles. (1: Tobermorite; 2: Portlandite; 3: Gypsum; 4: Quartz; 5: Tricalcium aluminate).

**Figure 5 materials-13-03500-f005:**
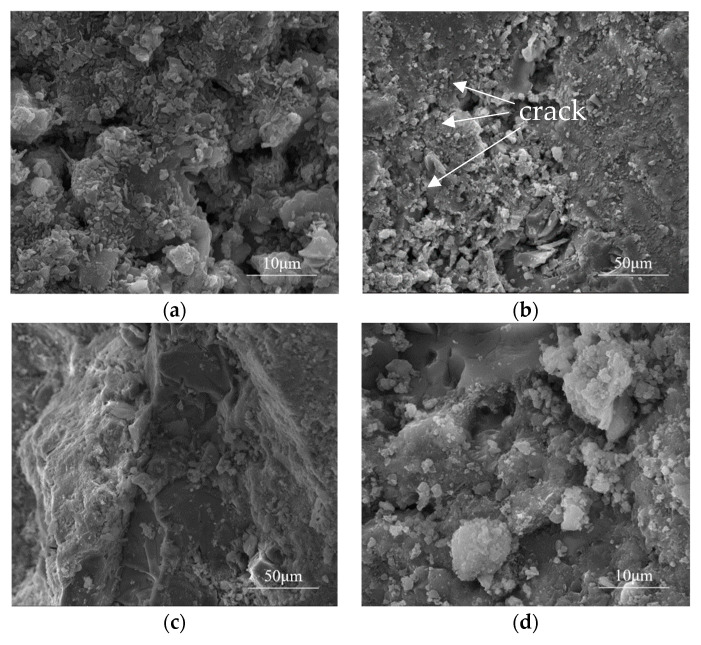
Scanning Electron Microscopy (SEM) diagrams of cement pastes and aggregate interfaces added with different admixtures after 300 sulfate dry–wet cycles; (**a**) OPC; (**b**) QP20; (**c**) SG20; (**d**) FA10.

**Figure 6 materials-13-03500-f006:**
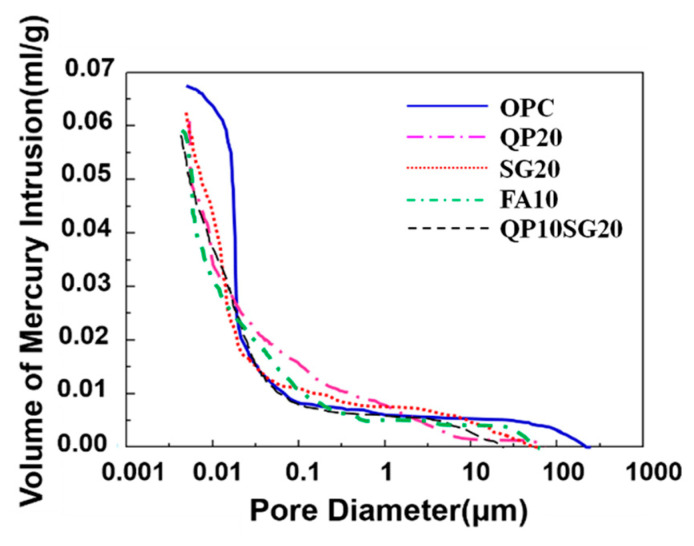
The relationship of volume of mercury intrusion and pore diameter.

**Figure 7 materials-13-03500-f007:**
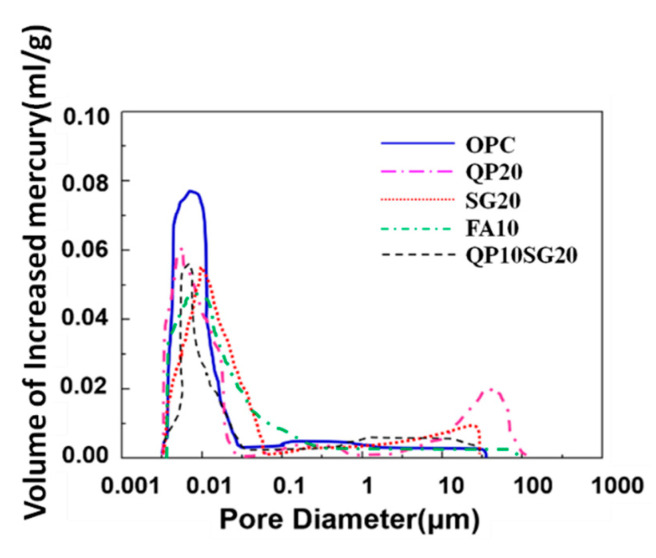
Pore distribution and pore diameter.

**Table 1 materials-13-03500-t001:** Physical and mechanical properties of ordinary Portland cement (OPC).

Fineness (wt. %)	Setting Time (min)		Flexural Strength (MPa)		Compressive Strength (MPa)	
	Initial Setting	Final Setting	3 d	28 d	3 d	28 d
1.8	100	160	4.8	7.9	22.7	47.0

**Table 2 materials-13-03500-t002:** Chemical composition of raw materials (wt. %).

Mineral Admixture	SiO_2_	Al_2_O_3_	Fe_2_O_3_	CaO	MgO	K_2_O	Na_2_O	SO_3_	Loss
OPC	22.62	6.11	3.69	57.96	2.16	0.98	0.17	3.00	2.6
QP	90.4	1.99	0.56	1.68	0.18	0.21	0.05	0.06	1.15
SG	34.43	14.64	1.08	40.78	6.78	0.31	0.33	2.12	0.87
FA	51.20	29.20	7.10	6.80	1.20	0.15	0.2	0.90	2.10

**Table 3 materials-13-03500-t003:** Coefficients $K_{n}$ of pre-stressed high-strength concrete (PHC) subjected to sulfate dry–wet cycles.

Sample		OPC	QP20	SG20	SG30	FA10	QP10SG20	QP10FA20
**K15**	R2(Mpa)/R1(Mpa)	96.2/94.3	95.9/93.1	104.7/99.7	102.6/96.8	84.6/82.9	103.4/99.4	82.3/79.1
	R2/R1	1.02	1.03	1.05	1.06	1.02	1.04	1.04
**K90**	R2(Mpa)/R1(Mpa)	106.8/94.5	103.2/93.0	114.2/100.2	110.0/96.5	92.8/82.9	115.5/99.6	86.9/79.0
	R2/R1	1.13	1.10	1.14	1.14	1.12	1.16	1.10
**K150**	R2(Mpa)/R1(Mpa)	106.5/95.1	102.3/93.0	113.5/100.4	109.2/97.1	90.4/82.2	112.8/99.8	90.9/79.7
	R2/R1	1.12	1.02	1.13	1.12	1.10	1.13	1.14
**K240**	R2(Mpa)/R1(Mpa)	70.5/95.3	49.4/92.6	77.7/100.9	78.0/97.5	64.2/83.4	75.9/99.9	56.9/79.0
	R2/R1	0.72	0.53	0.77	0.80	0.77	0.76	0.72
**K300**	R2(Mpa)/R1(Mpa)	44.7/95.2	-/93.7	53.2/100.3	53.6/97.5	44.3/83.5	55.0/98.3	42.4/81.6
	R2/R1	0.47	-	0.53	0.55	0.53	0.56	0.52

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
