# Peer review of "Effect of Mineral Admixtures on the Sulfate Resistance of High-Strength Piles Mortar"

_materials, 2020, doi:10.3390/ma13163500_

Round 1

Reviewer 1 Report

The article falls within the scope of the journal.

It is well structured and presented.

Some suggestions and comments can be found in the attached pdf.

Author Response

Dear Editors and Reviewers,

Thank you for your letter and reviewers’ comments on our manuscript entitled “Effect of Mineral Admixtures on the Sulfate Resistance of High-strength Piles Mortar” (ID: materials-880273). Those comments are valuable and very helpful to the revision and improvement of the paper, as well as the important guiding significance to our researches. We have studied comments carefully and have made modified which we hope meet with approval.

The revised part is marked in red in the paper. The main modified in the paper and the response to the reviewer’s comments are shown in the attachment.

Reviewer 2 Report

The analysed results and research area of this article is very interesting and actual. I have some remarks for improvement the quality of this article.

Remarks:

Fig.2-6 should be in one Fig. and marks, as a, b, c... should be used. Also the photos should be in the same size.

The title of 3.3 should be corrected.

In Fig. 7 should be indentified all higher peaks.

Fig. 8. "Portlandrite"? These Figures must be carefully corrected by adding other minerals ettringite, more peaks of portlandite, gypsum, alite, belite, etc. In the text You analyse these minerals, but they aren't in Figures. Also, how You can speak about amount of minerals, this method is only qualitative rather than quantitative. You didn't use internal standard. 

It should be added equipment and methodology of MIP, XRD and SEM tests.

Fig. 9. Why C1, S2 are 2 photos and G1, F1 - 1? It should be the same.

Author Response

(The authors gave the same response as above.)

Round 2

Reviewer 1 Report

The authors responded to the comments of the first review.

Please on your Kn equation, write down what this means, sth like "The n in??represents the number of dry-wet cyclesfor sulfate corrosion resistance."

Author Response

I have revised the third paragraph of 2.3 in accordance with the reviewer’s recommendations.

"?? represents strength and corrosion resistance coefficient; The n in ?? represents the number of dry-wet cycles for sulfate corrosion resistance; R 1 represents the compressive strength of the standard cured samples; R 2 represents the compressive strength of the samples under dry-wet cycles.”
